# MATPROPXTRACTOR: GENERATE TO EXTRACT

**Aswathy Ajith, Marcus Schwarting, Zhi Hong, Kyle Chard & Ian Foster**
Department of Computer Science
University of Chicago
Chicago, IL 60637, USA
`{aswathy,marcus,hong,chard,foster}@uchicago.edu`

## ABSTRACT

The field of materials science has amassed a wealth of information about materials in text publications, however, such information is often confined within the publication. A lack of standardized structure and naming consistency preclude the information from being effectively utilized for research and discovery. We introduce MATPROPXTRACTOR, an extraction system that uses pre-trained large language models (LLMs) in a generative setting to extract materials and their properties as reported in the materials science literature. MatPropXtractor consists of a three-step pipeline that includes 1) a document selection tool to identify related articles, 2) a paragraph classifier to identify passages containing important materials properties, and 3) a property extractor exploiting in-context learning in GPT-3. MATPROPEXTRACTOR extracted 154 material-property pairs from five materials science papers. The extracted pairs were analyzed by an expert and obtained an average precision of 72.73% on paragraph classification and an average precision of 56.7% precision on material-property identification.

## 1 INTRODUCTION

Extracting materials and their properties from existing literature is an ongoing research challenge Tchoua et al. (2019). Unfortunately, automated extraction methods face unique challenges related to materials naming inconsistencies and lack sufficient human-annotated data. Previous work focuses on rule-based approaches Swain & Cole (2016) or requires large volumes of human-annotated data, thereby limiting generalizability to other domains Shetty et al. (2022); Weston et al. (2019). They also require significant preprocessing of the data, including text extraction from PDFs and paragraph segmentation to be done by the end-user, slowing down the widespread adoption of such models by the research community. We propose a new approach that leverages auto-regressive large language models (LLMs) to extract materials and their properties from a publication, using minimal human-annotated data. This framework can be extended to other domains with limited human intervention. We hope this method serves as a useful tool for materials scientists to efficiently extract materials and properties of interest from specific publications.

## 2 METHODOLOGY

This section describes the primary operations of MATPROPEXTRACTOR, illustrated in Figure 2 consisting of three steps: 1) document filtering, 2) paragraph chunking and classification, and 3) property extraction.

**Document Filtering**: We use a random subset (80%) of the abstracts of papers from PPPDB as documents representing the materials science domain. The abstracts are encoded using SBERT Reimers & Gurevych (2019) to obtain real-valued vector embeddings. A simple threshold-based model is leveraged to identify documents within the same domain. For a test document, we obtain the average cosine similarity of its abstract to those in the document pool. The document is classified as pertaining to materials science, if it has an average cosine similarity above 0.6. This threshold was empirically determined and is domain-dependent.

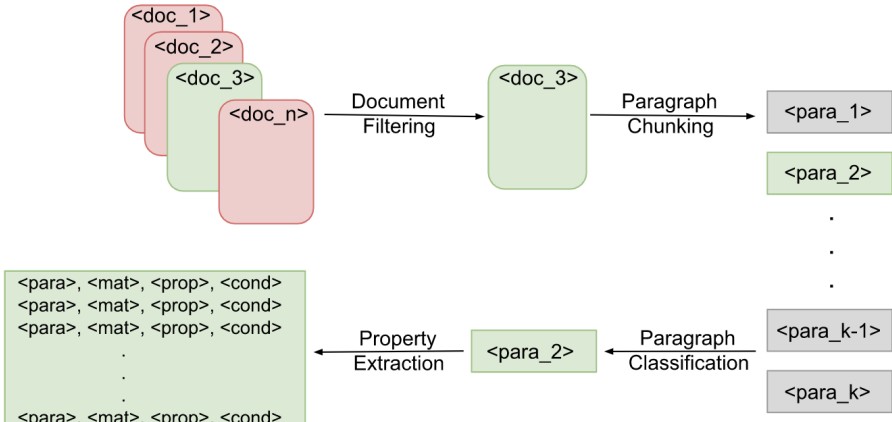

Figure 1: Extraction Pipeline of MATPROPXTRACTOR.

Table 1: An example of material-property pairs extracted from Zhou et al. (2010)

| Material | Property Name | Property Value |
|---|---|---|
| PMPCS 42 -b-PBLG 80 | columnar diameter | 1.58 nm |
| PMPCS 42 -b-PBLG 112 | columnar diameter | 1.56 nm |
| PMPCS 42 -b-PBLG 49 | d-spacing | 0.48 nm |
| PMPCS 42 -b-PBLG 19 | d-spacing | 0.48 nm |

**Paragraph Chunking and Classification**: We use the Grobid software GRO (2008–2023) to extract the text from PDFs and split them into paragraphs. Next, we randomly select a set of (four) paragraphs from the materials literature annotated as positive/negative examples depending on the presence of material-property pairs. We provide 2 positive and negative examples in the prompt.

**Property Extraction**: In order to extract material-property pairs from a given paragraph, we prompt GPT-3 Brown et al. (2020) model with a few examples that include the paragraph, followed by ⟨material, property name, property value⟩. The outputs from GPT-3 are then parsed to obtain tuples consisting of the paragraph, material, property name, property value, and additional constraints (e.g., experimental process used to obtain the property).

## 3 RESULTS

We obtain a total of 154 material-property pairs with MATPROPXTRACTORfrom a set of five papers (see Appendix for more details). The extractions were evaluated by a senior graduate student with a background in materials science. We report the precision of the extracted results here. The paragraph classification and the property extraction phases of the pipeline obtain an average precision of 72.73% and 56.7%, respectively. Our evaluation is limited to a small set of papers since it involves using the paid API services of an LLM such as GPT-3. We defer comparison to other systems to future work as there are no few-shot extraction pipelines or annotated datasets for low-resource domains like materials. However, the property extraction phase could be compared against a fine-tuned named entity recognition model, although this would not be a fair comparison.

## 4 LIMITATIONS

Paragraphs extracted by Grobid can be incorrectly segmented, propagating erroneous inputs to the subsequent steps of the pipeline. Grobid extractions with missing or incomplete mathematical quantities or symbols can cause inaccurate material-property pairs to be extracted. Finally, the sensitivity of LLMs to input prompts can limit the robustness of our methodology.

URM STATEMENT

The authors acknowledge that at least one key author of this work meets the URM criteria of ICLR 2023 Tiny Papers Track.

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

# A  APPENDIX

## A.1  SAMPLE OUTPUTS

Below we include selected sections of paragraphs across five papers, along with the identified materials and properties. Material names are highlighted in blue, properties in red, and values in green.

From Lee et al. (2003): "Styrene-butadiene diblock copolymer (denoted by S-B52) is the same material used in the previous study, where it was designated as S-B52. It contains 52 wt% of styrene as determined by the NMR technique, and its $M_n$ and $M_w$ are 25,000 and 26,000, respectively. The styrene-butadiene random copolymer (denoted by SBR60) containing approximately 60 wt% styrene was kindly synthesized for our use by Kumho Chemical Co., and its $M_w$ (by LS) and $M_w/M_n$ (by GPC) are 50,000 and 1.07, respectively."

From Xuan et al. (2004): "A previous description has shown that when symmetric P(S-b-MMA) is cast on a Si substrate with a SiO x surface layer, the PMMA preferentially segregates to the substrate while the PS segregates to the air interface due to the presence of asymmetric boundary condition. ... Furthermore, water contact angle (CA) measurement results show that the CA changes from 92° the surface of Figure 1a to P68° of the surface of Figure 1m. Since the water contact angle of PS homopolymer film is 90° and that of PMMA film is 66°, the changing of CA further suggests the migration of PMMA to the top surface."

From Basu et al. (2014): "A differential scanning calorimetric study also corroborated the findings obtained from DMA analysis (Figure ). The DSC glass transition temperature of pristine rubber is found at −24°C and, after treatment with ZnO and ZnCl$_2$, is shifted to -16 and +7 °C, respectively. Additionally, ZnO-treated samples showed a separate glass transition-like behavior at nearly 82°C. Thus, the nature of the DSC curves does not show melting-like behavior but rather glass transition-like behavior, resulting from the newly developed zinc-polymer in the system. The existence of high-temperature mechanical relaxation in DMA and the thermal transition in DSC are evoked by the glass transition of the newly formed network."

From Wibowo et al. (2014): "Polyion complexes (PICs) ... Here, the procedure to control PIC nanoarchitectures with various morphologies was established for the first time by careful tuning in the composition of PICs made from PEG-based block-ionomers with a varying amount of homoionomers as additive to modulate the PEG weight fraction ($f_{PEG}$) in the obtained PICs. Accordingly, the variation in $f_{PEG}$ from 12.1% to 6.5% induced vigorous transition in the microphase separated structures of PICs basically from micelle to lamella via cylindrical network. Notably, uniformed lamella with alternative layers of PEG and PIC domains was found at elevated temperature (70 °C), which, by lowering temperature, reversibly transformed to cylindrical PIC network apparently with connected aqueous channel in mesoscopic scale."

From Zhou et al. (2010): "In order to account for the self-assembly of the copolymers,... The 1D WAXD pattern of PMPCS 42-b-PBLG 49 exhibited rather similar profile to that of PMPCS 42-b-PBLG 19 for the $\Phi_N$ phase of PMPCS with a d-spacing of 1.56 nm, and the reflection for PBLG was also found at 4.48 nm -1 with a d-spacing of 1.40 nm, which corresponded to a columnar diameter of 1.62 nm. The appearance of ..."

## A.2   INPUT PROMPTS

### A.2.1   PARAGRAPH CLASSIFICATION

All annotated inputs were created by a non-expert human with limited domain knowledge. For the paragraph classification step, we use the following prompt as input to GPT-3:

"Does the given text contain material property information?

Text: Phase Structure and Self-Assembly in Bulk. DSC was used to study the thermal transitions of the diblock copolymers. Glass transitions were observed at  15 C for PBLG blocks and  130 C for PMPCS blocks during the first cooling and subsequent heating scans of the copolymers. PLM was used to observe birefringence of the copolymers. Sample films were cast from THF solutions by drying at room temperature. The liquid crystalline birefringence was observed at around 135 C. The texture of PMPCS42-b-PBLG19 is shown in Figure 2 as an example. The birefringence did not disappear when the sample was heated to 190 C, which was near the limit of thermal stability for PBLG, and subsequently cooled to room temperature, which implied that the samples formed their liquid crystalline phases at high temperatures (above 135 C) and these phases remained unchanged upon cooling.
Answer: yes

Text: Recently, we have designed and obtained a new kind of rod- rod diblock copolymer poly2,5-bis[(4-methoxyphenyl)oxycar- bonyl]styrene-b-poly($\gamma$-benzyl-L-glutamate) (PMPCS-b-PBLG). In this contribution, we describe here the synthesis of these block copolymers by combining atom transfer radical polymerization (ATRP) of 2,5-bis[4-methoxyphenyl]oxycarbonyl)styrene, ring-opening polymerization (ROP) of $\gamma$-benzyl-L-glutamate N-car- boxyanhydride, and a subsequent copper-catalyzed click che- mistry to form a series of rod-rod BCPs with different volume fractions of the two blocks. The phase behavior of the rod-rod BCPs with different volume fractions were investigated in this study, and hierarchical self-assembling structures on nanometer scale were analyzed by differential scanning calorimetry (DSC), polarized light microscopy (PLM), wide-angle X-ray diffraction (WAXD), and transmission electron microscopy (TEM) techni- ques. To the best of our knowledge, this is the first example of utilizing a facile and efficient click coupling reaction to prepare rod-rod diblock copolymers containing an R-helical polypeptide and a rigid vinylic polymer, although syntheses of rod-coil BCPs by click chemistry have been reported in literature.
Answer: no

Text: Characterization. The molar masses were determined with the combination of gel permeation chromatography (GPC), 1H NMR, and matrix-assisted laser desorption/ionization time-of- flight mass spectrometry (MALDI-TOF MS) measurements. GPC experiments were conducted on a Waters 2410 instrument equipped with a Waters 2410 RI detector and two Waters $\mu$-Styragel columns (103, 104 A  ), with THF or DMF as eluent (1.0 mL/min) in presence of LiBr (1 g/L). The calibration curve was obtained with linear polystyrenes as standards. 1H NMR spectra were obtained with a Bruker 400 MHz spectrometer. MALDI-TOF MS measurements were performed on a Bruker Autoflex high-resolution tandem mass spectrometer. Thermo- gravimetric analysis (TGA) was performed on a TA Q600 SDT instrument in nitrogen atmosphere. DSC examination was carried out on a TA Q100 DSC calorimeter in nitrogen atmo- sphere. PLM observation was performed on a Nikon DS-Ri1 microscope with an Instec HCS302 hot stage. One-dimensional (1D) WAXD experiments were carried out on a Philips X'Pert Pro diffractometer with a 3 kW ceramic tube as the X-ray source (Cu KR) and an X'celerator detector. Two-dimensional (2D) WAXD patterns were obtained using a Bruker D8Discover diffractometer with a GADDS as a 2D detector calibrated with silicon powder and silver behenate. The oriented films by mechanical shearing were mounted on the sample stage with the point-focused X-ray incident beam either parallel (X direction) or perpendicular (Y or Z direction) to the shear direction (X direction). The background scattering was recorded and sub- tracted from the sample patterns. TEM was used to investigate the microphase separation of samples on a Hitachi H-800 electron microscope. The solution-cast and ultramicrotomed sample films were stained by RuO4 vapor to enhance contrast.
Answer: no

Text: To further confirm the phase structure of the block co- polymers, 2D WAXD experiments were carried out on mechanically sheared samples. Parts a and b of Figure 5 show the 2D WAXD patterns taken with the X-ray beam along Y and Z directions at room temperature, respectively. In Figure 5a,

two pairs of strong diffraction arcs could be observed on the meridian in the low-angle region, which were attributed to the N phase of PMPCS and the H phase of PBLG, respectively. In Figure 5c, a set of Bragg reflections from the hexagonally ordered packing of PBLG at q = 4.7, 8.3, 9.8, and 12.6 nm-1 was observed with a scattering vector ratio of 1:31/2:41/2:71/2 in the integration at a 30-angle centered at the meridian shown in Figure 5a, which was consistent with the 1D WAXD results. The columnar dia- meter of PBLG helices was 1.54 nm. In Figure 5d, only a pair of diffraction arcs could be observed on the meridian, in which the reflection from the N phase of PMPCS was embedded in the first-order reflection of the H phase of PBLG as mentioned above in the 1D WAXD results. The 2D WAXD results indicated that the long axes of the PMPCS and PBLG rods were both aligned parallel to the X direction and to the lamellar normal. In other words, the PMPCS rods were aligned parallel to the PBLG rods. For the 2D WAXD patterns of the sample of PMPCS42-b-PBLG112, a set of Bragg reflections from the hexagonally ordered packing of PBLG at q = 4.8, 8.6, 10.1, and 12.7 nm-1 was also observed with a scattering vector ratio of 1:31/2:41/2:71/2 in the integra- tion at a 30-angle centered at the meridian shown in Figure 5d, in which the columnar diameter of the PBLG hexagon was 1.52 nm from the data above (shown in Figure 5f).
Answer: yes

Text: At high f PEG composition ( f PEG 12.1 and 11.1%), resulting PIC solution appeared to be transparent, and dynamic light scattering (DLS) results showed the formation of particles with sizes ranging from 4050 nm (Table S1, Supporting Information). Transmission electron microscopy (TEM) observations revealed the particles to be spherical with diameter %20 nm (Table 1, Figure 1a,b). As these features are consistent to polymeric micelles, thus the observed spherical particles is reasonable to assume as PIC micelles as reported previ- ously.12,14,18 Slight decrease in PEG weight fraction ( f PEG composition 10.0%) also gave transparent solution and the TEM observation showed spherical structure together with cylindrical structures (Figure 1c), where the former (18.1 ± 4.2 nm) with a diameter similar to the one obtained at f PEG = 12.1 and 11.1% (20.1 ± 2.9 and 20.0 ± 3.8 nm respectively) and the latter with a short axis of 12.4 ± 1.7 nm with substantially distributed long axis length (239.2 ± 91.0 nm; Table 1), implying shift from spherical polymeric micelles to cylindrical polymeric micelles with decrease of fPEG.
Answer: no'

## A.2.2 PROPERTY EXTRACTION

Extract materials and their properties from the text.

'Text: Figure 3a-c shows that carbonyl peak at 288.5 eV binding energy has increased gradually from zero, and the - peak for the phenyl ring around 292 eV has decreased gradually with increasing the PMMA selective solvent treatment time. This suggests the moving of PMMA to the top surface. Furthermore, water contact angle (CA) measurement results show that the CA changes from 92° of the surface of Figure 1a to 68° of the surface of Figure 1m. Since the water contact angle of PS homopolymer film is 90° and that of PMMA film is 66°, the changing of CA further suggests the migration of PMMA to the top surface.
Extraction:
⟨start⟩
material | property | value | constraints
PS | water contact angle | 90° |
PMMA | water contact angle | 60° |
⟨end⟩

Text: According to the above data from FTIR, 1D WAXD, and 2D WAXD experiments, the PBLG blocks could be assigned to a tightly packed hexagonal domain in the lamellar structure, and were considered to be packed next to the columnar nematic domain formed by the PMPCS blocks for sample PMPCS42-b-PBLG112. The proposed structure of the diblock copolymer is similar to the so-called HL or a cylinders-in-lamella morphology which has been reported on PBLG-b-polyglycine and PBLG-b-poly(L-lysine) peptide copolymers, and also in some other rod-coil systems.21,35,49 The PBLG segment had a hexagonal packing, with a columnar diameter of 1.56 nm for sample PMPCS42-b-PBLG112. Twice of the molecular lengths of the copolymers (47 nm for PMPCS42-b-PBLG80 and 56 nm for PMPCS42-b-PBLG112) on the basis of the molecular length simulation agreed well with the TEM observations, and the difference between the

layer distances (about 9 nm) of the two samples was close to twice of the increase in length (4.8 nm 2) of the PBLG block, in which the polypeptide segments adopted an 18/5 R-helical conformation and the PMPCS was presumed to have an extended chain conformation of the vinyl backbone. A bilayer structure appears more reasonable since the interdigitated lamellar structure has a maximal layer distance less than 30 nm by simulation. Therefore, we propose a stacked bilayer structure in the HL morphology for the self-assembly of PMPCS42-b-PBLG80 and PMPCS42-b-PBLG112.
Extraction:
⟨start⟩
material | property | value | constraints
PMPCS42-b-PBLG112 | columnar diameter | 1.56 nm | hexagonally packed PBLG segment
PMPCS42-b-PBLG80 | molecular length | 47 nm |
PMPCS42-b-PBLG112 | molecular length | 56 nm |
⟨end⟩

Text: Solvent Selectivity Effect. The miscibility between a polymer and a solvent is governed by the polymer-solvent interaction parameter PS (S ) solvent and P ) polymer). Using the Flory-Huggins criterion, the complete solvent-polymer miscibility can be realized when PS ¡ 0.5. The smaller the value is, the stronger the affinity between solvent and polymer is. According to the literature51-53 and the expression PS ) Vs[(dS - dP)2 + (pS - pP)2]/RT, where Vs is the molar volume of the solvent, R is the gas constant, d is the dispersion solubility parameter, and p is the polar solubility parameter, Chl-PMMAS ) 0.39, Chl-PS ) 0.45; Ace-PMMA ) 0.18, Ace-PS ) 1.1; Tol-PMMA ) 0.45, Tol-PS ) 0.34; and CS2-PMMA ) 1.2, CS2-PS ) 0.01.
Extraction:
⟨start⟩
material | property | value | constraints Chl-PMMA | polymer solvent interaction parameter | 0.39 |
Chl-PS | polymer solvent interaction parameter | 0.45 |
Ace-PMMA | polymer solvent interaction parameter | 0.18 |
Ace-PS | polymer solvent interaction parameter | 1.1 |
Tol-PMMA | polymer solvent interaction parameter | 0.45 |
Tol-PS | polymer solvent interaction parameter | 0.34 |
CS2-PMMA | polymer solvent interaction parameter | 1.2 |
CS2-PS | polymer solvent interaction parameter | 0.01 |
⟨end⟩

Text: We bring forward a mechanism of solvent vapor annealing for this case. As cast, since PMMA has a favorable interaction with the native silicon oxide, the surface tension of PS is comparatively lower (PS ) 40.7 mJ/m2, PMMA ) 41.1 mJ/m2);33,59 thus, this system favors asymmetric morphologies that have PS-rich layer at the free surface and PMMA-rich layer at the subvapor good for PMMA, the films is covered with solvent molecules. In this vapor environment, the boundary condition is different from that in the air or in a vacuum. Both Si substrate and solvent vapor layer at film surface absorb PMMA preferentially; for PMMA-solvent the interaction parameter is less than the PS-solvent interaction parameter (PMMA-S ¡ PS-S). Solvent vapor molecules have a stronger tendency to attract PMMA than PS to maximize the PMMA-solvent contacts, so PMMA is pulled toward the film surface (Figure 8b). With the increasing of vapor treatment time, more and more PMMA occupied the film surface. The confirmation of the migration of the PMMA block to the up surface is obtained by XPS (Figures 3 and 4) and water contact angle measurement. When treated in chloroform vapor for certain time (i.e., 60 h), the polymer chains are frozen by moving out from the solvent vapor environment and dried quickly. Ordered hexagonally packed structure is obtained to minimize the system total energy in case of PMMA uprising (Figure 8c,d). After an extended duration of the treatment further, surface-perpendicular morphology appears (Figure 8e,f). For a long enough treatment time (120 h), due to the interaction of PMMA between the Si substrate and the solvent, the PMMA block takes the stretched conformation and the PS block takes the collapse conformation. Therefore, PMMA occupies the upper surface and the partial substrate regions (Figure 8g,h).
Extraction:
⟨start⟩
material | property | value | constraints
PS | surface tension | 40.7 mJ/m2 |

PMMA | surface tension | 41.1 mJ/m2 |
PMMA-solvent | interaction parameter | < PS-S |
⟨end⟩

