# OpenReview forum: "MatPropXtractor: Generate to Extract"
_ICLR.cc/2023/TinyPapers — Submitted to Tiny Papers @ ICLR 2023_

### Official Review · Reviewer_fbNA · 2023-03-31

**Confidence:** 4

**Summary Of Contributions:**

The paper proposes a new extraction system called MATPROPXTRACTOR that uses pre-trained large language models (LLMs) in a generative setting to extract materials and their properties as reported in the materials science literature

**Rating:**

Great Start (GS): a submission which meets some of the reviewing criteria but has room for improvement

**Strengths And Weaknesses:**

Strength

- The demonstration and the results look promising and could be useful for extracting materials and properties.

Weakness

- There is no comparison with previous or existing methods. This would be highly valuable in judging the effectiveness of the new approach.

**Suggested Changes:**

It will beneficial to include a quantitative comparison with previous or existing methods in the paper

---

### Official Review · Reviewer_XhU9 · 2023-04-03

**Confidence:** 5

**Summary Of Contributions:**

This paper proposed MatPropXtractor. It first pre-processes the text data from documents. And then use manually labeled examples as prompts to the GPT-3. Finally, it uses GPT-3 for material extraction.

**Rating:**

Needs Clarification (NC): a submission which does not meet the reviewing criteria and needs clarification for its described problem or solution

**Strengths And Weaknesses:**

## Strengths:
### 1. The paper is clearly written and easy to read.
### 2. Figure 1 is a good illustration of the pipeline.


## Weaknesses:
### 1. There is no novelty in this paper.
The proposed MatPropXtractor simply does data preprocessing and applies the GPT-3 to the material extraction task. There is no model trained, and it is only a direct application of GPT-3 with some labeled prompts.

### 2. The paper writing is not good enough. It is a rash submission.
There are many grammar mistakes and typos. For example, in the Abstract, 'preclude' should be 'precludes', 'materials properties' should be 'material properties', and there should have a full stop before the last sentence.


**Suggested Changes:**

1. Try different ways of prompting and summarize the best strategy. This would increase the contribution of the paper.

---

### Meta-Review · Area_Chair_mE8c · 2023-04-06

**Recommendation:** Invite to archive
**Confidence:** 5

**Metareview:**

**Summary**
* The paper introduces a new extraction system named as MatPropXtractor. This tool uses pre-trained large language models (LLMs) in a generative setting to extract materials from material science literatures and report their properties.

**Strengths**
* The work is valuable in the sense that the method could be used to extract information from material science literature.


**Weakness**
* The paper lacks details about who are the experts and how many of them participated in manually labeling the properties from the material literature. This is highly important to validate the results and get a sense of the performance of the LLMs in doing this particular task.
* Discussing how other LLMs compared to GPT-3 would be valuable to compare the results.


**Summary:**

The paper introduces a new extraction system named as MatPropXtractor. This tool uses pre-trained large language models (LLMs) in a generative setting to extract materials from material science literatures and report their properties. Strength, the work is important to extract knowledge from material science literatures. Weakness: the paper lacks details.

**Comments And Feedback To The Authors:**

Very interesting application in an important problem. It would be great if you could add details even in the appendix for other researchers to know the results and build up on the application.

**Reason For Not Giving A Higher Recommendation:**

Although the paper proposes an interesting application for LLMs, it requires revisions to provide details to claim their performance compared to experts. In order to validate the results, it would be important to include who the experts are and how many of them participated in the study.

Although I understand the 2-page limit could be a constraint, they could add it on the appended, since this is valuable information to fulfill the completeness of the work.

**Reason For Not Giving A Lower Recommendation:**

N/A

---

### Decision · Program_Chairs · 2023-04-07

Invite to archive